

# A retrospective prognostic evaluation using unsupervised learning in the treatment of COVID-19 patients with hypertension treated with ACEI/ARB drugs

Liye Ge[1], Yongjun Meng[1], Weina Ma[1] and Junyu Mu[2]

[1] Jiading District Central Hospital Affiliated Shanghai University of Medicine and Health Sciences, Shanghai, China

[2] Nanjing Medical University, Nanjing, China

## ABSTRACT

**Introduction**. This study aimed to evaluate the prognosis of patients with COVID-19 and hypertension who were treated with angiotensin-converting enzyme inhibitor (ACEI)/angiotensin receptor B (ARB) drugs and to identify key features affecting patient prognosis using an unsupervised learning method.

**Methods**. A large-scale clinical dataset, including patient information, medical history, and laboratory test results, was collected. Two hundred patients with COVID-19 and hypertension were included. After cluster analysis, patients were divided into good and poor prognosis groups. The unsupervised learning method was used to evaluate clinical characteristics and prognosis, and patients were divided into different prognosis groups. The improved wild dog optimization algorithm (IDOA) was used for feature selection and cluster analysis, followed by the IDOA-k-means algorithm. The impact of ACEI/ARB drugs on patient prognosis and key characteristics affecting patient prognosis were also analysed.

**Results**. Key features related to prognosis included baseline information and laboratory test results, while clinical symptoms and imaging results had low predictive power. The top six important features were age, hypertension grade, MuLBSTA, ACEI/ARB, NT-proBNP, and high-sensitivity troponin I. These features were consistent with the results of the unsupervised prediction model. A visualization system was developed based on these key features.

**Conclusion**. Using unsupervised learning and the improved k-means algorithm, this study accurately analysed the prognosis of patients with COVID-19 and hypertension. The use of ACEI/ARB drugs was found to be a protective factor for poor clinical prognosis. Unsupervised learning methods can be used to differentiate patient populations and assess treatment effects. This study identified important features affecting patient prognosis and developed a visualization system with clinical significance for prognosis assessment and treatment decision-making.

Corresponding authors
Weina Ma, maweina26@163.com
Junyu Mu, junyumu@163.com

## INTRODUCTION

The ongoing COVID-19 pandemic is posing an enormous challenge to the global healthcare system. Older adults, those with preexisting respiratory or cardiovascular disease, and those with diabetes and high blood pressure are at increased risk of serious complications and death (*Fang, Karakiulakis & Roth, 2020*; *Asselah et al., 2021*). Available data suggest that hypertension is one of the most common comorbidities in COVID-19 patients and is associated with a more severe disease course and higher mortality (*Schiffrin et al., 2020*; *Gasmi et al., 2021*). Among COVID-19 patients, those with high blood pressure may experience more severe disease and a greater risk of death (*Castiglione & Droppa, 2022*).

However, data on hypertensive patients and their use of antihypertensive drugs are very limited, and the efficacy and impact of drug therapy remain controversial (*Gallo, Calvez & Savoia, 2022*). Since SARS-CoV-2 infects target cells *via* the receptor angiotensin-converting enzyme 2 (ACE2), there is controversy as to whether angiotensin-converting enzyme inhibitor (ACEI) and angiotensin II receptor blocker (ARB) may be associated with hypertension and worse outcomes in patients with COVID-19 (*Bhandari et al., 2016*; *Zhao et al., 2021*). Although certain adverse events, such as drug ineffectiveness, have been reported for ACE class drugs, such as benazepril and captopril, it is important to consider the possibility of adverse events in our patient sample (*She et al., 2021*). However, it is worth noting that the specific ACEI/ARB drugs prescribed were not explicitly mentioned (*Majd et al., 2024*). In fact, animal models and human studies have shown that ACEI/ARB may increase lung ACE2 levels (*Rico-Mesa, White & Anderson, 2020*; *Ma et al., 2021*).

In view of the complex effects of ACEI/ARB drugs, the limited clinical trial data available do not support the combination of COVID-19 with differential application of renin-angiotensin-aldosterone system (RAS) inhibitors in hypertensive patients (*Erdine et al., 2006*; *Bozkurt, Kovacs & Harrington, 2020*), and individualized treatment should be based on the patient's clinical manifestations and hemodynamic status (*De Backer et al., 2022*). Recent research reports have shown a lower risk of death from COVID-19 infection in patients treated with ACEI/ARB than in hypertensive patients not treated with ACEI/ARB (*Kumar et al., 2022*). In addition, there is insufficient evidence that ACEI/ARB affect the risk of COVID-19, and no deleterious effects of ACEI/ARB on hospitalization or in-hospital death have been established (*Cheng et al., 2009*; *Li et al., 2021*). Hence, the appropriateness of using ACEI/ARB antihypertensive drugs in patients with COVID-19 and hypertension still requires further evaluation. The impact of their use on the progression of COVID-19 and whether they increase the risk of infection have attracted the attention of experts and patients with cardiovascular disease at home and abroad (*Ye & Liu, 2020*; *Kai & Kai, 2020*; *Bauer et al., 2021*).

In past studies, conclusions about the role and safety of ACEI/ARB in the treatment of COVID-19 were inconsistent (*Zhang et al., 2024*). Some studies suggest that these drugs may increase patients' risk of SARS-CoV-2 infection and lead to worsening of the disease (*Trougakos et al., 2021*; *Angeli et al., 2022*). However, it has also been suggested that ACEI/ARB may play a role in protecting the lungs from viral attack by increasing soluble ACE2 levels (*Kumar & Al Khodor, 2020*; *Martínez-Gómez et al., 2022*). To fully

understand the efficacy and prognostic impact of ACEI/ARB in patients with COVID-19 and hypertension, this study used unsupervised learning methods to analyse and evaluate large-scale clinical data.

Unsupervised learning is a machine learning method whose goal is to discover hidden structures and patterns from data without the need for prespecified labels or target variables (*Huyan et al., 2022*; *Marcon et al., 2022*). Unlike supervised learning, unsupervised learning is not constrained by prelabelled data and can explore information in the dataset more flexibly (*Ma et al., 2022*). Unsupervised learning can help us understand the effect and impact of ACEI and ARB drugs in the treatment of COVID-19 complicated with hypertension by allowing us to comprehensively consider the clinical characteristics, treatment options and prognoses of patients (*Ju et al., 2023*). In this study, an unsupervised learning method was applied to the clinical data of patients with COVID-19 and hypertension to explore the effect and potential mechanism of ACEI and ARB drug treatment. With this data-driven approach, we can evaluate treatment effects more comprehensively and provide more accurate guidance for clinical practice. This approach will facilitate the development of personalized treatment strategies to improve the prognosis and survival of patients with COVID-19 and hypertension.

This study aimed to observe the clinical characteristics of patients with COVID-19 and hypertension during the course of treatment with ACEI/ARB drugs and the impact of these drugs on the outcome of the disease to provide a basis for medical teams to make decisions. This research will help to further understand the impact of high blood pressure and antihypertensive drugs on COVID-19 and provide doctors with more accurate treatment options. The contributions and innovations of this study can be described as follows: (1) In terms of the data, this study used large-scale clinical data covering the characteristics of clinical symptoms, laboratory tests, and imaging examinations of patients with COVID-19 and hypertension. By analysing these data, more comprehensive and accurate information can be obtained to evaluate the effect and impact of ACEI/ARB drug treatment. (2) Methodologically, this study used unsupervised learning methods to discover similarities and differences between patients through techniques such as clustering and dimensionality reduction algorithms and revealed the potential effects and mechanisms of ACEI/ARB drug therapy. This data-driven approach can better mine hidden information in the data and provide comprehensive assessment and guidance. (3) In terms of application, we developed a visualization system for the prognostic assessment of patients with COVID-19 and hypertension. Through unsupervised learning prognostic assessment, the effect of ACEI/ARB drug treatment can be accurately evaluated with a low threshold, and a scientific basis can be provided for formulating individualized treatment strategies. This approach will help improve patient prognosis and survival and optimize medical care.

## METHODS

We utilized an unsupervised model to classify patient outcomes. Subsequently, key feature difference analysis and feature importance ranking were performed on the good prognosis

and poor prognosis groups and then verified *via* unsupervised analysis. To facilitate prognostic assessment, we also developed a visualization system. The technical flowchart of this study is shown in Fig. 1, which includes the data collection, feature extraction, improved algorithm and simultaneous optimization, feature selection and cluster analysis, and model training and evaluation steps.

## Data collection and feature extraction

This study was approved by the Ethics Committee of Shanghai City Jiading District Hospital (2022-K10). Informed consent was obtained from all individual participants included in the study. This was a single-center, retrospective study of patients with COVID-19 from Tongjia Designated Hospital in Jiading District, Shanghai city. With the approval of the Ethics Committee of Shanghai City Jiading District Central Hospital, informed consent was waived. Patients with COVID-19 and hypertension were selected for the study. The inclusion period ranged from April 7, 2022 to May 19, 2022. A total of 200 patients (96 males and 104 females) were included. The final follow-up date was June 21, 2021.

The inclusion criteria for patients were as follows: (1) refer to the "Diagnosis and Treatment Plan for Novel Coronavirus Pneumonia (Trial Version 10)" (*Nicola et al., 2020*) and "Guidelines for the Prevention and Treatment of Hypertension in China (2018 Edition)" (*Théry et al., 2018*). According to the relevant diagnostic criteria in the study, (2) a comprehensive evaluation of the epidemiological history, clinical symptoms, laboratory indicators and imaging findings of the patients was performed; the exclusion criteria were as follows: (1) aged less than 18 years; (2) had SARS-CoV-2 asymptomatic infection; (3) had advanced malignant tumors and needed radiotherapy and chemotherapy; (4) had severe organ failure at admission; and (5) had complete organ failure.

We collected demographic information from two groups of patients, namely, clinical symptoms, diagnosis and classification, auxiliary inspection and clinical outcome.

## Improved algorithm and synchronous optimization

The swarm intelligence optimization algorithm can provide a good foundation for feature selection and clustering analysis (*Hanrahan, 2011*; *Li et al., 2022*). The principle of the improved wild dog optimization algorithm (IDOA) is briefly introduced as follows. The name of our improved algorithm is IDOA-K-Means. First, the IDOA uses the cubic chaotic mapping strategy to initialize the population, and the chaotic mapping can be expressed as

$$x_{n+1} = \rho x_n(1 - x_n^2). \tag{1}$$

Here, $x_n$ corresponds to the numerical value of the nth dimension. To enhance the traversal, we set the initial value to $x_0 = 0.3$ and the parameter $\rho = 2.593$. In chaotic mapping, the selection of parameter values needs to be determined based on specific requirements and system characteristics. For cubic chaotic maps, the parameter $\rho$ affects the stability and periodicity of the system. When the parameter $\rho$ is 2.593, the cubic chaotic mapping has the best traversal performance, which means that the system can better explore various points in the state space. The purpose of setting the initial value to 0.3 is to

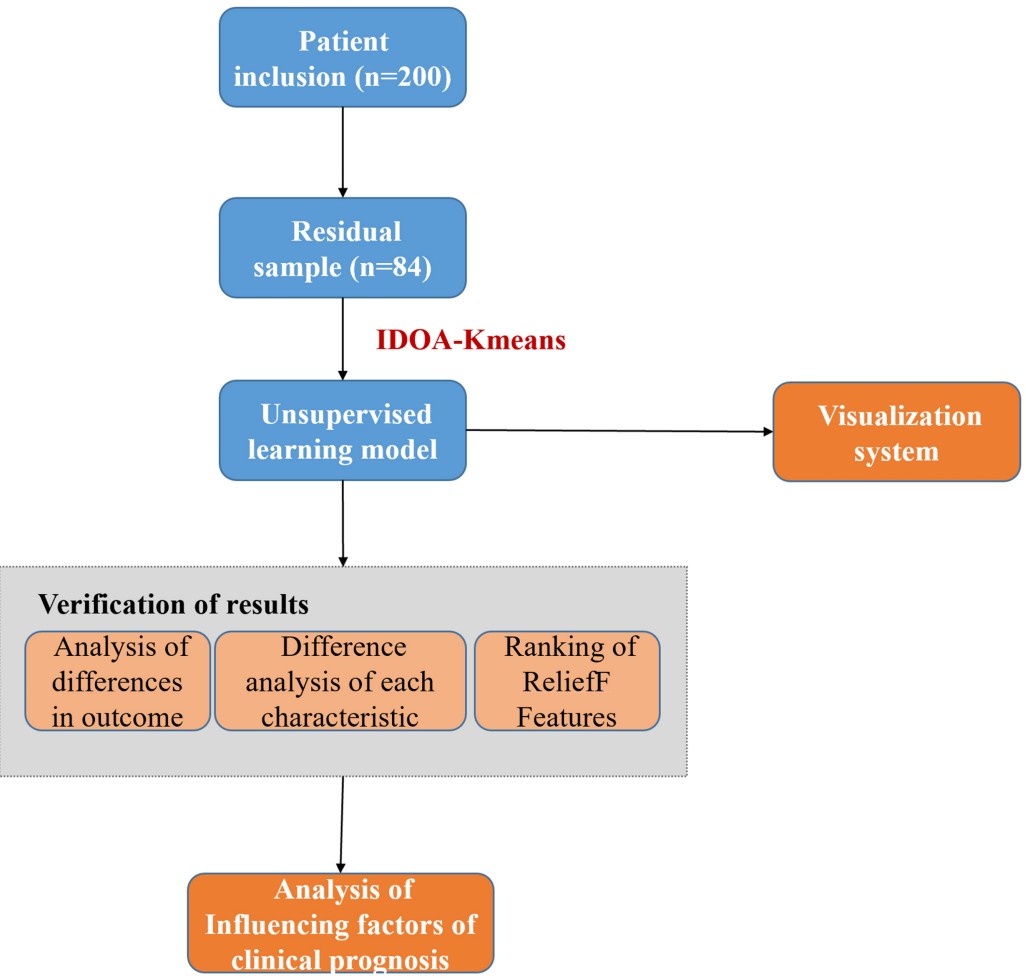

**Figure 1** **The technical flowchart of this study.** (1) Data collection: First, we collected the required data from appropriate sources, which may include demographic information, clinical indicators, laboratory test results, and other content. (2) Feature extraction: Next, we process the collected data and extract features related to the research purpose. (3) Improved algorithm and synchronous optimization: To improve the initial point selection dependency of the traditional k-means algorithm, we introduce the swarm intelligence algorithm and conduct synchronous optimization to achieve the task of feature filtering and initial point optimization. (4) Feature selection and cluster analysis: After optimization, we further conducted feature selection to identify features that had a significant impact on the prognosis. Next, we use the improved k-means algorithm to cluster the data and divide the samples into different clusters. (5) Model training and evaluation: Based on clustering analysis, we use the selected features to train the prediction model and evaluate the model to evaluate its performance and predictive ability. The cases (80 cases) that meet the inclusion criteria after screening of COVID-19 patients (200 cases) are called residual samples. Residual samples refer to the leftover samples after the initial analysis or processing steps have been completed.

ensure that the values generated by cubic chaotic mapping fall between [0,1]. This initial value selection takes into account the randomness of the initial state and the rationality of the mapping results, allowing the algorithm to better explore potential solution spaces. Through such an initialization method, the blindness of population initialization can be overcome so that the population can better cover the search space.

Second, the IDOA uses dimensionwise Gaussian variation for the optimal individual. Specifically, for each dimension j, the optimal position is changed by Gaussian mutation, which can be expressed as

$$Xbestnew(j) = w * Xbest(j) + randn * Xbest(j). \qquad (2)$$

In the process of dimensionwise Gaussian mutation, an inertia weight w is used to balance the influence of the new solution and the old solution. Xbest represents the best position before updating, and Xbestnew represents the new position after Gaussian mutation.

Finally, the IDOA uses a greedy strategy to update the fitness; that is, it uses a greedy strategy to retain the optimal solution, which can be expressed as

$$Xbest = \begin{cases} Xbestnew, if \ f(Xbestnew) < f(Xbest) \\ Xbest, else \end{cases}. \qquad (3)$$

This strategy aims to improve the ability of the optimal individual to jump out of the local optimal solution, thereby enhancing the global search ability of the algorithm. This means that if the fitness of the new solution is better, update the optimal solution to the new solution; otherwise, keep the original solution unchanged. Through such a strategy, the best solution can be obtained, and the performance of the algorithm can be improved.

Hence, by introducing dimensionwise Gaussian mutation and a greedy strategy, the IDOA is beneficial for overcoming the local optimal problem of traditional optimization algorithms and improving the global search ability of the algorithm. The core idea of the IDOA is to increase the diversity of the population through the mutation strategy to better search for the optimal solution. Using the 23 common test function to evaluate the performance of the IDOA, as shown in Fig. 2, the results reveal that the IDOA is superior to the control method in terms of convergence speed and the ability to obtain the global optimal solution.

## Feature selection and cluster analysis

After successfully constructing the IDOA, we use the global optimization ability of the IDOA to complete the two key steps of feature selection and optimal clustering center point optimization at the same time.

The Euclidean distance is used to calculate the clustering between the sample points and the cluster center; that is, the sum of squared errors (SSE) is used as the objective function for optimization, and the formula can be expressed as follows:

$$SSE = \sum_{i=1}^{k} \sum_{X_m \in C_i} \|X_m - C_i\|^2. \qquad (4)$$

Here, $C_i$ represents the cluster center point, and $X_m$ represents each sample data point. The cluster center point is updated through each iteration, and the update rule calculates the new center point coordinates according to the category to which the sample points belong. The smaller the $SSE$ is, the better the clustering effect, and the clustering center

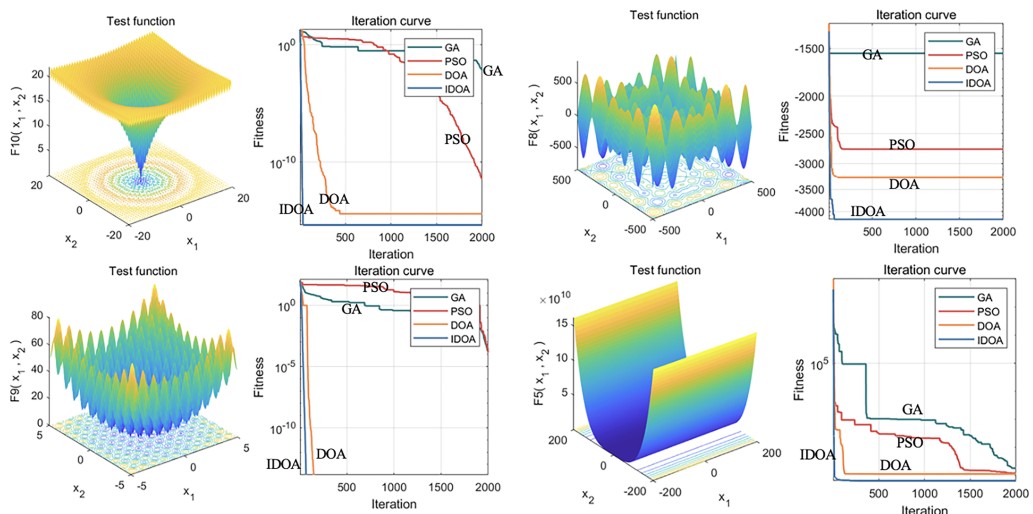

**Figure 2 Performance analysis of the IDOA on 23 test functions.** This figure shows the results of 23 common test functions used in the performance evaluation of the improved IDOA. The evaluation results indicate that the IDOA outperforms the control method in terms of convergence speed and global optimal solution acquisition ability. The convergence speed refers to the speed at which an algorithm reaches the global optimal solution from its initial state, while the ability to obtain the global optimal solution indicates whether the algorithm can find the optimal solution to the problem. The figure shows that the IDOA algorithm converges faster and has better global optimal solution acquisition ability for most test functions.

point is updated through each iteration $C_i$. The update rule can be expressed as follows:

$$C_i = \frac{1}{n} \sum_{m=1}^{n_i} X_m, i = 1, 2 \ldots, k. \tag{5}$$

Here, $n_i$ represents the number of data points in category $i$, and the cluster center point is updated through each iteration of the above formula until the cluster center does not change, indicating that the model converges effectively.

A flow chart of the synchronous optimization process is shown in Fig. 3. The initialization process determines the feature screening and the encoding method of the center point; the result of feature screening determines the dimension of the cluster center point; finally, the selected features and center point are merged, and optimization is performed on this basis. The detailed steps can be described as follows. First, in the initialization process, the feature screening part adopts 0–1 encoding, indicating whether the feature is retained, and the center point adopts real number encoding, indicating the coordinates of the center point. Second, 0 means to remove features, and 1 means to keep features. The dimensions of the center point are determined according to the number of features chosen. For example, three features are reserved in the above figure, so the first three real numbers are selected as the coordinates of the center point. Finally, the selected features and center point coordinates are merged, and the objective function is added to the merged dataset for optimization. Throughout the process, feature selection and central point optimization are carried out

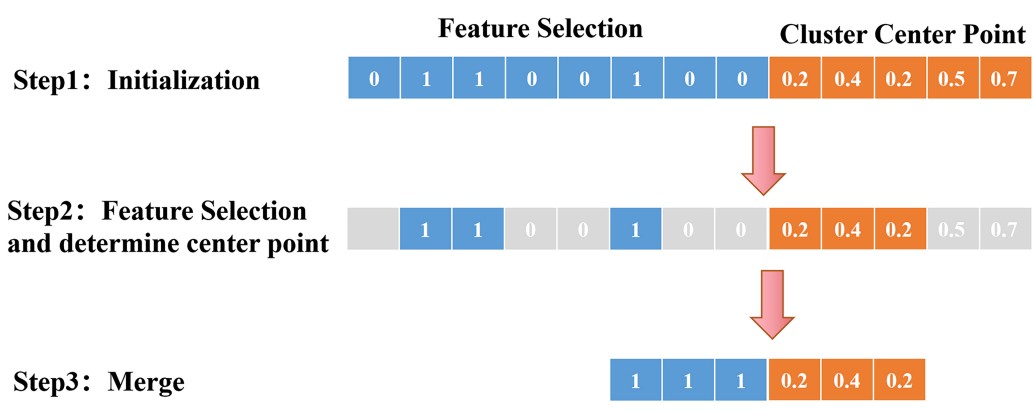

Figure 3 **Schematic diagram of synchronous optimization.** This process includes several key steps, including initialization, feature filtering, dimensionality determination of clustering center points, and merging and optimizing features and center points. From initialization to feature filtering, dimensionality determination of clustering center points, and then to the merging and optimization of features and center points, each step plays an important role, ultimately achieving optimization of clustering results.

simultaneously. By merging the datasets and inputting them into the objective function, the optimal features and the best clustering central points can be obtained.

## Model training and evaluation

To avoid the risk of overfitting and improve the stability of training, 50-fold cross-validation was applied to the training set. Subsequently, we evaluated the performance of the final model by selecting 80% of the total number of patients as the training set and the remaining 20% as the test set. This method can effectively avoid overfitting of the model on the training set and obtain more accurate prediction results on the test set.

The data were analysed using IBM SPSS (26.0) with a significance level set at 5%. Count data were compared using the t test (*Liu & Wang, 2021*), while categorical variables were compared using the $\chi^2$ test (*Aslam, 2021*).

## RESULTS

### Feature selection and cluster analysis results

This section compares the traditional k-means algorithm with the IDOA-K-means algorithm and provides a two-dimensional scatter plot and a comparison of the prognostic performance of k-means clustering analysis to demonstrate the differences before and after algorithm optimization.

First, in the cluster analysis, we screened out six characteristics that affected the differences between groups, age, hypertension grade, MuLBSTA (*Iijima et al., 2021a*), use of ACEI/ARB, NT-proBNP, and high-sensitivity troponin I. The MuLBSTA score (*Iijima et al., 2021b*) is a valuable and effective early warning model that has been developed to predict the mortality rate of patients with viral pneumonia. These features included multiple infiltrations, hypophocytosis, bacterial infection, smoking history, hyper tension, and age. Afterwards, to better visualize the clustering results, we applied principal component
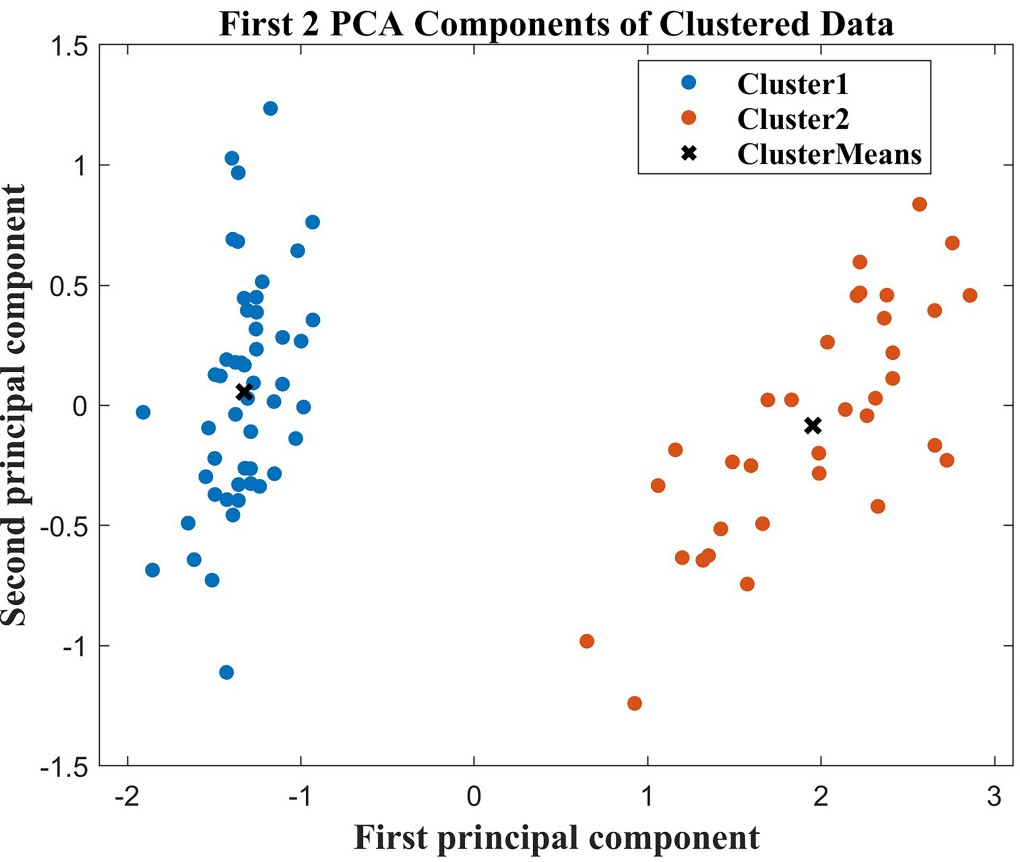

**Figure 4  Two-dimensional scatter plot after PCA dimensionality reduction.**

analysis (PCA) to reduce the dimensionality of these 6 features. In this dimensionality-reduced feature space, we draw a two-dimensional scatter plot, as shown in Fig. 4. The results of the scatter plot revealed that the clustering effect was good, and there was clear differentiation between the two groups of patients with good and poor prognoses. This shows that the application of the IDOA-k-means algorithm to this dataset is successful and that the algorithm can be used to classify patients effectively and identify distinguishing features.

However, we also conducted a prognostic performance analysis on the two groups processed by clustering analysis. The results before optimization are shown in Table 1, and the results after optimization are shown in Table 2. The results before improvement showed that k-means clustered the data into two categories, but there was no significant difference in the prognosis between the two groups of patients, which was not sufficient. The improvement results showed that the hospitalization days and nucleic acid conversion time in group 2 were greater than those in group 1, while the nucleic acid positivity rate (27.03%) in group 2 was greater than that in group 1. The differences in these indicators were statistically significant ($P < 0.05$). In addition, we also observed that there were no

**Table 1** Cluster analysis comparing the prognostic performance of the two groups of patients (before optimization).

| Group | | Group 1 | Group 2 | $t$ ($\chi^2$) | $P$ |
|---|---|---|---|---|---|
| No | | 53 | 31 | | |
| The number of days in hospital | (x ± s, d) | 10.79 ± 4.65 | 11.97 ± 5.54 | −1.041 | 0.301 |
| Nucleic acid negative time | (x ± s, d) | 12.72 ± 4.84 | 11.35 ± 5.47 | 1.187 | 0.239 |
| Nucleic acid Fuyang | [n(%)] | 5(9.43) | 5(16.13) | −0.836 | 0.361 |
| Die | [n(%)] | 1(1.89) | 1(3.23) | −0.151 | 0.698 |

Notes.
(x ± s) represents the mean and standard deviation, and d represents the number of days; this table represents the results before optimization.

**Table 2** Difference analysis results of the baseline information of the two groups of patients.

| Variable | Good prognosis ($n = 47$) | Poor prognosis ($n = 37$) | $t$ ($\chi 2$) | $P$ |
|---|---|---|---|---|
| Age (x ± s, years old) | 63.11 ± 13.72 | 74.78 ± 13.03 | −3.959 | <0.001 |
| Gender[n(%)] | | | (0.820) | 0.365 |
| male | 25 (60.98) | 16 (39.02) | | |
| female | 22 (51.16) | 21 (48.84) | (12.189) | <0.001 |
| Use ACEI/ARB[n(%)] | | | | |
| yes | 32 (74.42) | 11 (25.58) | | |
| no | 15 (36.59) | 26 (63.41) | | |
| Hypertension grade [n(%)] | | | (6.225) | 0.044 |
| Level 1 | 21 (75.00) | 7 (25.00) | | |
| level 2 | 18 (47.37) | 20 (52.63) | | |
| Level 3 | 8 (44.44) | 10 (55.56) | | |
| Types of COVID-19[n(%)] | | | (3.952) | 0.047 |
| light and normal | 47 (58.02) | 34 (41.98) | | |
| heavy | 0 (0) | 3 (100) | | |
| MuLBSTA (x ± s, points) | 5.64 ± 2.87 | 7.27 ± 3.45 | −2.365 | 0.020 |

Notes.
(x ±s) represents the mean and standard deviation, and d represents the number of days; this table represents the results after optimization.

deaths in group 1, while there were 2 deaths in group 2. Based on these observations, we classified group 1 as the good prognosis group and group 2 as the poor prognosis group.

## Difference analysis of key characteristics between the two groups of patients

This section analyses the differences in key characteristics of the two groups of patients from the aspects of baseline information, clinical symptoms, laboratory tests, and imaging examinations and then selects the key characteristics that affect the differences between the two groups, as shown in Supplementary Material S1.

First, for the baseline data, we observed that there were significant differences between the two groups in age, ACEI/ARB drug use, hypertension grade, new crown type and MuLBSTA ($P < 0.05$). These features are closely related to patient prognosis. Moreover, the clinical symptom indices exhibited minor discrepancies between the two patient groups

and did not demonstrate statistical significance ($P > 0.05$). Hence, clinical symptom indicators were not the key factors in predicting the prognosis of patients in this study. Third, regarding the laboratory test indicators C-reactive protein (CRP), interleukin-6 (IL-6), lactate dehydrogenase, D-2 polymer, oxygenation index, procalcitonin (PCT), aspartate aminotransferase (AST), creatinine, NT-proBNP, and hypersensitivity, there were statistically significant differences in lymphocyte counts between patients in the two groups. Additionally, there were significant differences in troponin I and cycle threshold of COVID-19 nucleic acid detection ($P < 0.05$). These laboratory test results may be highly important for predicting patient prognosis. Finally, in terms of imaging examination indices, there was no significant difference in the results of X-ray examination between the two groups ($P > 0.05$), which suggested that X-ray examination is useful for evaluating patient prognosis.

## Ranking and verification of feature importance

According to the results of the 17 features with statistical significance in the feature difference analysis, we used the ReliefF algorithm (*Zhang, Ding & Li, 2008*; *Liu, Chen & Huang, 2023*) to further rank the importance of these features, as shown in Fig. 5. In the ranking of the importance of features, the top six features were age, hypertension grade, MuLBSTA, use of ACEI/ARB, NT-proBNP, and high-sensitivity troponin I. The importance ranking results of these features are consistent with the results of unsupervised prediction model screening features, as demonstrated in the first part of the Results section ("Feature selection and cluster analysis results"). This further verified the correlation of these features with the prognosis of patients with new crowns. The ranking of the importance of these features can provide a reference for clinicians to help them more accurately assess the prognosis of patients and take corresponding treatment measures.

To verify the importance of these selected features, we combined the six features screened out by the model into a multifactor logistic regression model, as shown in Table 3. To address categorical variables, we turned them into dummy variables. The outcome variable was defined as the occurrence or absence of an adverse clinical outcome, where 0 indicates a good prognosis and 1 indicates a poor prognosis. After stepwise screening, we found that age (OR = 1.055, $P = 0.009$) and ACEI use (OR = 0.302, $P = 0.020$) were two significant independent variables.

## Visualization system construction

The aforementioned studies have successfully identified key features that influence patient outcomes. However, in clinical practice, the changes in these characteristics are intricate, and it is difficult to intuitively determine the prognostic risk. Existing artificial intelligence methods have high barriers to popularization and application, requiring clinicians to have high programming skills and extensive literature knowledge, which makes them difficult to promote and use in a large number of hospitals. To solve this problem, this article innovatively constructs a practical visualization system that is built on the basis of selected key features and has the advantages of intuition, convenience and practicability. During the application of the visualization system, the user needs to input only the specific values of

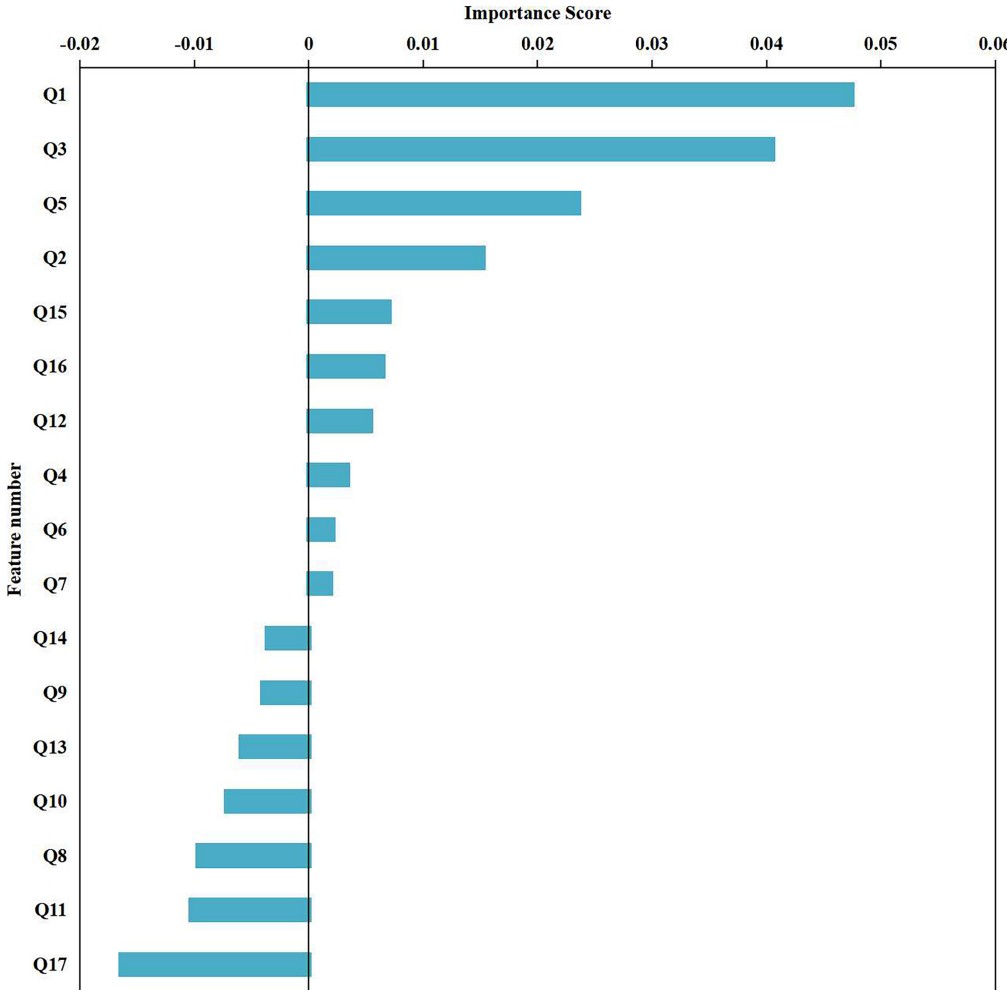

**Figure 5  Ranking of feature importance.** The 17 statistically significant characteristics in the feature difference analysis: age, whether to use ACEI/ARB, hypertension classification, COVID-19 classification, mulbsta score, lymphocytes, CRP, IL-6, lactate dehydrogenase, D-2 polymer, oxygenation, PCT, AST, creatinine, NT proBNP, hypersensitive sarcocalcin I, gene O were numbered Q1-Q17 respectively, and the importance of the features was sorted using the ReliefF algorithm.

the six key features in the "baseline information" column, and the system can automatically calculate the prognosis level and provide targeted suggestions. Figure 6 shows an example of an application for prognostic assessment. In the process of clinical diagnosis and treatment, this system helps to quickly screen for prognostic risk and take timely targeted measures to address high-risk patients. The construction of this system helps patients achieve timely clinical intervention, thereby reducing the risk of adverse outcomes, which has important practical significance and application value.

The code, software version, and parameters for "feature selection and clustering analysis" are available in GitHub: https://github.com/gelihua/IDOAKmeans. The difference analysis of key characteristics between two groups of patients was conducted using SPSS 26.0. Use MATLAB R2022a's built-in Train Classification Models in Classification Learner App

**Table 3   Multivariate logistic regression (stepwise method) results.**

| Variable | Regression coefficients | standard error | z | Wald χ² | P | OR | 95% CI of OR |
|---|---|---|---|---|---|---|---|
| Age | 0.053 | 0.020 | 2.627 | 6.901 | 0.009 | 1.055 | 1.014~1.098 |
| Using ACEI/ARB | −1.196 | 0.512 | −2.335 | 5.454 | 0.020 | 0.302 | 0.111~0.825 |
| Intercept | −3.355 | 1.517 | −2.211 | 4.887 | 0.027 | 0.035 | 0.002~0.684 |

**Notes.**
CI of OR represents the confidence interval of the Ods ratio (OR) obtained using chi square test with binary variables.

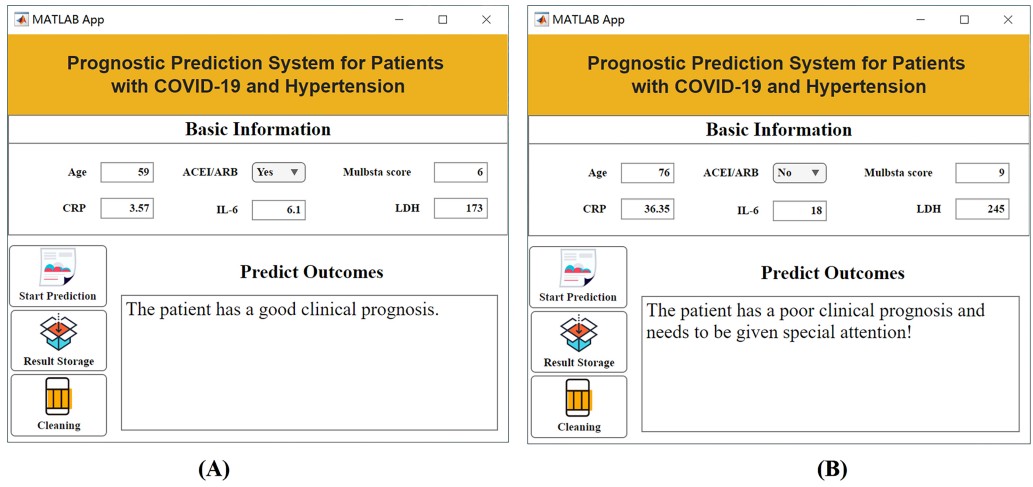

**Figure 6   Visualization system interface.** (A) Example of a patient with a good prognosis; (B) example of a patient with a poor prognosis.

for visualization of feature importance sorting and validation. The functional parameter selection is FeatureSelection ReliefF.

The code, software version, and parameters of the "visualization system" can be found on GitHub: https://github.com/gelihua/IDOAKmeans. This software is developed based on MATLAB R2022a and utilizes the APP Designer function to design the original *. mlapp file. On this basis, MATLAB's built-in MATLAB Runtime compiler is used to compile the *. mlapp file into a *. exe file that can be executed independently of the MATLAB environment. As long as the software is pre installed on a computer with MATLAB Runtime, it can be run, successfully reducing the running environment requirements of the software and improving portability. (1) Hardware requirements: Processor main frequency: 2Ghz and above; memory: 2GB or above. (2) Software requirements. System: Windows 10 and above (64-bit system); operating environment: MATLAB R2022a and above. MATLAB Runtime version 9.12 and above.

# DISCUSSION

ACEI/ARB are two commonly used antihypertensive drugs that regulate the RAS for blood pressure regulation. The receptor protein ACE2 of SARS-CoV-2 is also a component of the
RAS. The use of ACEI/ARB increases ACE2 expression, which may increase the severity of COVID-19 (*Li et al., 2021*). On the other hand, ACE2 itself has anti-inflammatory effects, suggesting a potential protective role in the progression of COVID-19. In addition, ACEI/ARB is beneficial for improving the prognosis of pneumonia patients, which in turn effectively assists in the treatment of COVID-19. Some scholars express concerns that the use of ACEI/ARB may increase the risk of COVID-19 viral infection and even lead to the development of severe conditions (*Xue et al., 2020*; *Ma et al., 2021*). However, other scholars hold the opposite view, stating that COVID-19 infection triggers the downregulation of the RAS (*Cheng et al., 2009*; *Kumar et al., 2022*). This results in activation of the RAS, which is one of the important mechanisms causing lung damage.

To resolve the above controversy, we explored the ability of unsupervised learning to predict the prognosis of hypertensive patients with COVID-19 following ACEI/ARB therapy. The key finding of this study was the improvement in outcomes observed in hypertensive patients with COVID-19 treated with ACEI/ARB. This finding suggested a potential protective role of ACEI/ARB in respiratory infections. The mechanism underlying this protective effect is thought to involve the upregulation of the ACE2 receptor, which has been shown to play a role in the entry of SARS-CoV-2 into host cells (*Angeli et al., 2022*). By blocking ACE2 receptors, ACEI/ARB may reduce the viral load and prevent severe lung injury (*Martínez-Gómez et al., 2022*; *Kumar et al., 2022*). In practical applications, doctors and patients need to comprehensively consider individual conditions, disease severity, and other relevant factors. Moreover, it is necessary to pay close attention to the latest research results and professional guidelines to make more accurate decisions. During the epidemic, regarding the use of ACEI drugs, the potential risks and benefits should be fully weighed and fully discussed and communicated with doctors.

The indicators related to the clinical prognosis of patients included the length of hospitalization, the duration of negative nucleic acid conversion, the duration of positive nucleic acid recovery and the likelihood of death. However, there is no clear definition of clinical prognosis using hospitalization time or nucleic acid negativity time. In addition, due to the low incidence of nucleic acid positivity and death (11.9% [10/84] and 2.3% [2/3], respectively), the sample size is limited, and it is difficult to perform accurate analysis *via* traditional methods (*Xue et al., 2020*). To solve this problem, we adopt the strategy of unsupervised learning and use the improved k-means algorithm to divide the samples into two categories with the largest difference. By validating the clustering results, we were able to more accurately analyse the factors affecting the prognostic outcome. Further multivariate analysis revealed that, for patients with COVID-19 and hypertension, the use of ACEI/ARB is a protective factor and can reduce the risk of adverse clinical outcomes. These findings could guide clinicians in more accurately assessing the prognostic risk of patients and formulating corresponding treatment plans. In addition, this method based on unsupervised learning can not only be applied to patients with COVID-19 and hypertension but also be extended to prognosis prediction research for other diseases. We choose not to use traditional supervised machine learning methods because our research problem has complex nonlinear relationships and a large number of features, which traditional supervised learning methods may find difficult to capture. In contrast, we

choose to use deep learning methods because they perform better in handling large-scale datasets and complex pattern recognition. Through deep learning, we can better mine hidden information and potential patterns in data, thereby achieving better predictive performance.

The ranking results from supervised validation are consistent with our previous results using unsupervised predictive models for feature screening, as shown in the first part of the Results section titled "Feature selection and cluster analysis results". With this unsupervised prediction model, we found that these features have a significant impact on predicting patient prognosis. First, age is a very important feature that plays a key role in predicting patient prognosis. The older the patient, the less favorable the prognosis may be. Second, hypertension grade and the MuLBSTA score are also closely related to patient prognosis. The higher the hypertension grade and the higher the MuLBSTA score are, the worse the prognosis may be. In addition, the use of ACEI/ARB, NT-proBNP levels and high-sensitivity troponin I levels is also an important feature for predicting patient prognosis. The use of ACEI/ARB drugs may have a positive impact on patient prognosis, and the levels of NT-proBNP and high-sensitivity troponin I may be related to cardiac function and the inflammatory response and have certain predictive value for prognosis.

Overall, the ranking results of the importance of these features are consistent with the results of the unsupervised prediction model screening features, further verifying the relevance of these features to the prognosis of patients with new crowns. The ranking of the importance of these features can provide a reference for clinicians to help them more accurately assess the prognosis of patients and take corresponding treatment measures. Hence, younger age and the use of ACEI/ARB drugs can help reduce the incidence of adverse clinical outcomes. These results have important value for clinicians in predicting the prognosis of patients and formulating treatment plans. In addition, the application of unsupervised learning has broad potential in the medical field to help reveal the underlying mechanisms of diseases, predict patient risks, and optimize treatment options. By combining clinical data and machine learning techniques, we can better understand and respond to global epidemics such as COVID-19 and provide patients with better treatment recommendations.

However, it should be noted that there are several limitations in this study. First, there may be selection bias due to the limited sample size. Second, there may be certain errors in the collection and analysis of clinical data. Hence, future research needs to further expand the sample size and incorporate more precise data collection and analysis methods to improve the accuracy and reliability of the predictive model.

## CONCLUSION

In summary, based on an unsupervised learning strategy and the improved k-means algorithm, we successfully achieved an accurate analysis of the prognosis of patients with COVID-19 and hypertension. The results of the study clearly indicated that the use of ACEI/ARB drugs has a protective effect on the clinical prognosis of patients. With an unsupervised learning approach, we are able to efficiently differentiate between different

patient populations and evaluate the effects of drug treatments. In addition, we identified important features that affect patient outcomes and developed targeted visualization applications that provide clinicians with valuable tools. The results of this study have important clinical significance for guiding the prognostic evaluation and drug treatment decision-making of patients with COVID-19 and hypertension. Our research provides physicians with more information and bases so that they can more accurately assess patients' prognostic risk and develop individualized treatment plans. According to the specific characteristics and conditions of the patient, the doctor can better judge whether to administer ACEI/ARB drug treatment and can monitor the patient's response and adverse reactions to adjust the treatment plan in a timely manner.

### Funding

This work was supported by the Shanghai Key Specialty Project of Clinical Pharmacy (No.YXZDZK-01), the Nature Science Foundation of Jiading District, the Shanghai (No.JDKW-2021-0043) and the Shanghai University of Medicine and Health Sciences Clinical Research Centre for Metabolic Vascular Diseases Project (No.20MC2020004). The funders had no role in study design, data collection and analysis, decision to publish, or preparation of the manuscript.

### Grant Disclosures

The following grant information was disclosed by the authors:
Shanghai Key Specialty Project of Clinical Pharmacy: No. YXZDZK-01.
Nature Science Foundation of Jiading District, Shanghai: No. JDKW-2021-0043.
Shanghai University of Medicine and Health Sciences Clinical Research Centre for Metabolic Vascular Diseases Project: No. 20MC2020004.

### Competing Interests

The authors declare there are no competing interests.

### Author Contributions

- Liye Ge conceived and designed the experiments, performed the experiments, analyzed the data, prepared figures and/or tables, authored or reviewed drafts of the article, and approved the final draft.
- Yongjun Meng conceived and designed the experiments, performed the experiments, analyzed the data, prepared figures and/or tables, authored or reviewed drafts of the article, and approved the final draft.
- Weina Ma conceived and designed the experiments, performed the experiments, analyzed the data, prepared figures and/or tables, authored or reviewed drafts of the article, and approved the final draft.
- Junyu Mu conceived and designed the experiments, performed the experiments, analyzed the data, prepared figures and/or tables, authored or reviewed drafts of the article, and approved the final draft.

## Data Availability

The data and code is available at GtiHub and Zenodo:

- https://github.com/gelihua/IDOAKmeans.
- gelihua. (2023). gelihua/IDOAKmeans: V1.0.0 (master). Zenodo. https://doi.org/10.5281/zenodo.10207627.

## Supplemental Information

Supplemental information for this article can be found online at http://dx.doi.org/10.7717/peerj.17340#supplemental-information.

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
