# Peer review of "A retrospective prognostic evaluation using unsupervised learning in the treatment of COVID-19 patients with hypertension treated with ACEI/ARB drugs"

_PeerJ, doi:10.7717/peerj.17340_

## Round 0.1 · original submission · Major Revisions

Major Comments:

Methods:

1. From the method section it is not clear how the authors performed this analysis. For example, 1) programming language used? 2) Did the authors performed the analysis using any software or did they performed the analysis using in-house scripts?
2. The authors also mentioning the construction of the visualization system in abstract, introduction and results. However, there is no methodology described for visualization system construction in the method section. Also, throughout the manuscript it is not clear how the readers and the users are going to access this visualization system? Is there a website where this visualization system is hosted? Which programming language were used to create this system? From figure 6, I can guess that the authors have used MATLAB to generate this system. However, only providing the interface figure is not enough. A interface demo would be more useful.
3. Provide the step -by step analysis, code, software versions and parameter used for the analysis in a supplementary file or provide a GitHub repo with a detailed readme. The authors do not have to provide their data (if they don’t want to share it publicly)- a small test dataset should be OK.
4. Section 2: “Improved Algorithm and Synchronous Optimization” : Too much information included to introduce the IDOA method. The authors should consider moving some of this information in the supplementary text, including Figure 2. Also provide a high resolution image for Figure 2; It is very hard to review it in the current form.
5. The authors should considering defining why they choose the initial value x0=0.3 and the parameters ρ=2.593 for formula given in line 163.
6. Line 186: The authors should provide more details about the 23A common test function.

Results:

1. Line 237-241 and 243-252: please move them to the method section
2. Line 287: It is not clear which “previous result conclusion” are the authors mentioning here. Provide the proper citations.

Discussion:

1. Line 338-340: The authors states that “The key finding of this ……treated with ACEIs/ARBs”. However, these finding are not describe in the results section at all. Please include the description of the key findings in the result section.
2. The authors also needs to cite proper literature to support their statements in the discussion section. One such example is the statement given in line the 366-367

General comment: The authors have used many novel technologies, but there is no proper detailed description. The authors could consider to include a detailed description of previous studies regarding this in the introduction and description section.

Minor comments:

Line 27: MuLBSTA; provide full form.
Line 58: RASS; provide full form.
Line 58: COVID-19-19 should be COVID-19.
Line 66: “After use on COVID-19” please rephrase the sentence.
Line 117: “Informed consent Informed consent” remove the duplicate words.
Line 133: Provide full form of HIS

·

Basic reporting

1) Why ACEIs and ARBs are together? Why are they not documented separately, which means two different variables?
2) Insufficient literature search.
More previous studies should be documented regarding why hypertensive medicines such as ACEIs and ARBs are crucial for mortality prediction.
3) Better to say patients with COVID-19 instead of COVID-19 patients.
4) Would it be possible to release the raw dataset and coding?
5) Which programming/s is used here? (Matlab is used in Figure 6).

Experimental design

1) Lines 119-121: Explain why Informed Consent was waived.
2) Line 186: Please explain the 23A Common Test more.
3) Line 243: Please explain the IDOA-K-Means algorithm to your audience.
4) Line 287: Need citation for this line: “with previous research conclusions.”
5) Lines 290 to 292: Please name a few other studies regarding ReliefF algorithm.

Validity of the findings

1) There may be a separate section for statistical results. A more statistical description is needed in the writing section. In addition, a comparison of results is needed between the statistical and modeling (unsupervised learning method).

2) Lines 329 to 331: There is an explanation of ACEIs, and why is there no explanation of ARBs since both variables are together?

3) In the discussion section, we need to compare with the other study results to know whether there are any agreements or disagreements of the findings.

4) Why are there below 50 citations? Please explain that no more studies on the subject exist or an incomplete literature search. Please include literature on other diseases (e.g., infectious), too.

5) The unsupervised learning method used here is validated in the other study. Should it be explained or compared with some other validated method if you build up this (novel techniques not previously used to analyze similar data)?

Additional comments

Please check the following study. This study presents a supervised learning model where Age is the most important feature, but ARBs and ACEIs are below the 10 most important features.

Datta, D., George Dalmida, S., Martinez, L., Newman, D., Hashemi, J., Khoshgoftaar, T. M., Shorten, C., Sareli, C., & Eckardt, P. (2023). Using machine learning to identify patient characteristics to predict mortality of in-patients with COVID-19 in South Florida. Frontiers in Digital Health, 5, 1193467. https://doi.org/10.3389/fdgth.2023.1193467

Figures:
1) There is a detailed description needed for each figure.
2) Figure 2: A detailed description is needed. Pictures are not legible.
3) More detailed information is needed for Figure 3. The picture is partly visible.
4) How the ranking of features was done in Figure 5.
5) Figure 5: No detailed description

Typos/Not Clear:
1) Line 58: COVID-19-19
2) Line 66: After the use of COVID-19

Reviewer 2 ·

Basic reporting

Ge et al. used unsupervised learning to cluster patients with COVID-19 and hypertension and identified clinical parameters that were able to distinguish between those with poor and good prognoses.

1. The authors used clear, unambiguous and professional English
2. The literature references and context provided is sufficient.
3. The overall article structure is professional. However, I do have a few minor comments that could improve the quality of the tables and figures:
• Figure 1: instead of writing “inclusion and exclusion”, state the reason for exclusion, followed by (n=).
• Figure 2: the resolution is poor and I was unable to read the wording on the figure. If unable to improve the resolution, rather summarize key findings
• Table 1: difficult to read. Consider adding Clusters (groups) as headings and features as rows
• Table 2: Age and MuLBSTA score are continuous features, they shouldn’t have chi-squared results .
• Table 2: the p-value and chi-squared test results for ACE/ARB are not in the right place. Please modify
• Table 4: It looks like the authors are reporting the median and the (p25th -p75th ) interquartile range. However, the results do not show a range, but show what looks like a standard deviation. Consider removing [M(IQR)] next to each feature and state this in the footnote of the table.
• Table 5: instead of “have” and “none” consider using “yes”and “no”
• Table 5: what do authors mean?: “X-rays are used instead of imaging examinations?” A chest x-ray is an imaging examination. Please rephrase
• Table 6: Can authors elaborate on how they created the logistic regression model? Which variable did they use as the outcome or independent variable?
• Table 6: what is the significance of “project.”
4. The results are relevant to the hypothesis.

Experimental design

1. Original primary research within the aims and scope of the journal.
2. The research question is well-defined, relevant & meaningful. It is stated how research fills an identified knowledge gap.
3. Rigorous investigation performed to a high technical & ethical standard.
4. Methods described with sufficient detail & information to replicate.
5. Few comments:
• There are many unsupervised learning methods. What is the rationale for using the improved wild dog optimization algorithm? How did the authors decide on this algorithm? Indeed, the wild dog algorithm uses Gaussian variation, but there are other unsupervised algorithms that are based on partitioning, hierarchy and density of data points in the higher dimensional space.
• Since it was not possible to review Figure 2. How did the authors test the performance of the unsupervised algorithm? What were the performance metrics on the training and test dataset?
• State how features with a non-normal distribution were handled.
• What is the protocol for treating hypertension in the hospital? How do clinicians decide on whom to treat with ACEI/ARB vs beta-blockers?
• Were patients included in the study taking other anti-hypertensives or ACEI/ARB only?
• Data collection: please rectify lines 117-118.

Validity of the findings

1. Some underlying data have been provided; they are robust, statistically sound, & controlled.
2. Conclusions are well stated, linked to the original research question & limited to supporting results.
3. In terms of feature importance, which features correspond to Q17, Q11, Q8 and Q10. These features are important but have a negative impact on the model.
4. I am not certain if authors can confidently conclude that ACE/ARB use affects prognosis, considering that the weight of ACE/ARB (feature Q2) ranked number 4. Age, hypertension grade and MuLBSTA had higher weights.
5. What were the baseline systolic and diastolic pressures in each cluster?
6. Is MuLBSTA a score? How is it calculated? Please clarify in the methods section.

Additional comments

None

---

## Round 0.2 · Major Revisions

Thank you for the revised manuscript. The authors have answered most of my previous comments. However, there are a few of my previous comments that still need to be answered. Please see below:

Previous comments not answered:

Line 133: Provide full form of HIS (Line 148 in the revised manuscript)
In the supplementary material, the authors did provide the visualization system set up instruction but did not provide the source code. Please provide the *. mlapp file and*. exe files (as supplementary materials) that can be executed independently of the MATLAB environment.
Provide the step -by step analysis, code, software versions and parameter used for the analysis in a supplementary file or provide a GitHub repo with a detailed readme. Please provide this information in the revised manuscript instead of describing this in the rebuttal letter.
The authors should considering defining why they choose the initial value x0=0.3 and the parameters ρ=2.593 for formula given in line 163 (line 168 in the revised manuscript). Instead of rebuttal letter please provide this information in the supplementary file.
Line 27: MuLBSTA; provide full form.
The authors also needs to cite proper literature to support their statements in the discussion section. One such example is the statement given in line the 366-367 (lines 377-378 in the revised manuscript)

Revised manuscript comments:

Line 24 (Abstract); Move this line to the method section of the abstract.
Lines 60-61; Provide reference to support the statement “individualized treatment should be based on the patient's clinical manifestations and hemodynamic status”
Line 103; “This paper develops a visualization system” should be “ We have developed a visualization system”.
Lines 119-128; Merge the note with the Fig 1 legend. The new Fig1 legend should be: “The technical route of this study: 1) Data collection ………….”.
Line 148: Provide full form of HIS
Lines 147-149: As per these lines, the authors have utilized “HIS electronic medical management information system and the Rimy laboratory management system to analyse the patient demographic information”.
However, in line 149, it is not clear what is “(including gender, age, blood pressure level)” means here? Are these the features of the Rimy laboratory management system? Or are these the feature association with patients?. Please correct this line accordingly.
In lines 147-158; Please correct the English in this sentence. The authors may consider dividing the big sentence into smaller parts if that helps.

Lines 162-163: Please rephrase the sentence.
Line 167: Define the formula; For instance, what does x and n represent?
Lines 191-197; Merge the note with the Fig 2 legend. The new Fig 2 legend should be: “Performance analysis of the IDOA on 23 test functions. This figure shows the results of ……”
Line 232-236: Merge the note with the Fig 3 legend.
Line 244-245: Rephrase the sentence.
Lines 259-260; “two groups of patients have obvious distinction” – describe which two groups of patients.
Table1 and table2 both have same title “Cluster analysis comparing the prognostic performance of the two groups of patients”. Also, in the footnote of Table 1; describe what is (XS, d).
Table 3-6 should be supplementary tables.
Table 3, 4 and 6: In the footnote of the table describe n(%).
Table 5: what is [M(IQR)]? Describe this in the table footnote.
Table 7: What is “CI of OR:? Describe this in the table footnote
Lines 289-290: full form : CRP, IL-6, PCT, AST
Lines 301-302; “with the results of unsupervised prediction model screening features”. Provide the reference for this result.
Lines 337-338; No need to provide the full form of ACEIs and ARBs as authors have already provided this in the introduction section.
Three different abbreviation used for renin-angiotensin-aldosterone system. These are : in lines 59-60; renin-angiotensin-aldosterone system (RAAS), in lines 338-339; renin-angiotensin-aldosterone system (RAS) and in lines 347-348; renin-angiotensin system (RAS).Please correct the term accordingly.
Line 351: After the sentence “…… patients with COVID-19.” Include the sentence “The key finding of this study is the improved outcomes observed in hypertensive patients with COVID-19 treated with ACEIs/ARBs.”.
Line 377-400: Cite references to support the statements.
Please be consistent with the usage of ACEIs/ARBs or ACEI/ARB throughout the manuscript.

·

Basic reporting

They made changes /corrected all the reviews.

Experimental design

They made changes /corrected all the reviews.

Validity of the findings

They made changes /corrected all the reviews.

Additional comments

Does the study compare with other drugs treating hypertension that do not belong to the ACEI/ARB family? If it is not; why?

Reviewer 2 ·

Basic reporting

No further comments

Experimental design

No further comments

Validity of the findings

No further comments

Additional comments

No further comments

·

Basic reporting

The article aligns with the journal's guidelines, ensuring data deidentification and adherence to ethical standards. The response to the previous review has been duly addressed. Furthermore, thorough checks have been conducted on the figures and tables within the manuscript. Overall, I appreciate the authors' efforts in presenting a clear and understandable research layout. The figures are pertinent and appropriately labeled, enhancing comprehension. However, the manuscript still requires revisions.

Experimental design

Line 55: Rephrase sentence starting with “On the one hand”.
ACE class drug for example, Benazepril and Captopril has certain FDA adverse events reported including drug ineffectiveness (https://drugcentral.org/drugcard/299?q=Benazepril%20). Authors probably need to address the possibility of the adverse events reported in the patient-sample. However, which exact drugs from the ACE/ADRB class were prescribed is also unclear.

Validity of the findings

Line 158-159: please rephrase.
Provide justification as to why the IDOA and the k-means algorithms are the most suitable for the research problem among other ML algorithms. You could use the type of the target variable to justify this.
Line 245: mention the variables which were normally distributed.
Line 254: Rephrase sentence starting with “On the one hand”.
Line 282: Tables 3-6.
The manuscript contains excessive repetition of phrases like "On the other hand" and "On the one hand," which may distract from the scientific clarity needed for a scholarly audience. Overall, the document requires substantial revisions to enhance its presentation and make it more suitable for a scientific article. The current language usage might hinder its appeal to a scientific audience due to its lack of clarity and precision. Elaborating on the points without relying excessively on these phrases would significantly improve the manuscript's scientific readability and impact.

Reviewer 4 ·

Basic reporting

The authors addressed all reviewers' comments on basic reporting.

Note that all subfigures under figure 2 are too small. Please enlarge them for better readability. In addition, it would also be beneficial to improve the clarity of the images.

Experimental design

The authors addressed all reviewers' comments on experimental design.

Validity of the findings

The authors addressed all reviewers' comments on the validity of the findings.

Note that for the clarification on MuLBSTA score, please move the explanation in the rebuttal letter to the methods section in the paper.

---

## Round 0.3 · Major Revisions

Thank you for the revised manuscript. Please see the comments below:

Major comments: The following are my previous comments that the authors haven't answered.

1) "Please be consistent with the usage of ACEIs/ARBs or ACEI/ARB throughout the manuscript". For example, lines 69, 351, 355, 364-365, and 384 still have ACEIs/ARBs

2)  Instead of only a rebuttal letter, (a) please include the description of (XS,d) in the footnote of Table 1 and Table 2. b) Modifies the legend of Table 1 and Table 2 accordingly so that it is clear that these tables represent the results before optimization in case of Table 1, and the results after optimization for Table 2.
3) In response to my comment (i.e., "In the supplementary material, the authors did provide the visualization system set up instruction but did not provide the source code. Please provide the *. mlapp file and*. exe files (as supplementary materials) that can be executed independently of the MATLAB environment."), the authors have mentioned in the rebuttal letter that "For the convenience of readers and other researchers, we have provided the executable file you requested as supplementary material; please refer to GitHub for additional details". 

However, I did not find any executable files in the supplementary material. There is no GitHub link given in the supplementary material and the main manuscript for additional details.

Please note that to reproduce the results this is very important that the authors provide their source code used for the analysis and executable files for the visualization system. 

4) Provide the step-by-step analysis, code, software versions, and parameters used for the analysis in a supplementary file or provide a GitHub repo with a detailed readme. Please provide this information in the revised manuscript instead of describing this in the rebuttal letter.

The authors have provided this information in the supplementary materials only for the visualization system. The authors need to provide the codes, software versions, and parameters used for the analysis of "Feature selection and cluster analysis", "Difference analysis of key characteristics between the two groups of patients", and "Ranking and verification of feature importance" as well. Please make sure that this information is included in the next revision.  

New comments- Revised manuscript comments are given below:

1) Lines 59-60; Please cite the reference to support the statement "However, it is worth noting that the specific ACEI/ARB drugs prescribed were not explicitly mentioned".
2) Line 75; Provide the full form of CVD.
3) Lines 77-78; Cite references to support the statement "In past studies, conclusions about the role and safety of ACEIs and ARBs in the treatment of COVID-19 were inconsistent".
4) Line 82; " To more fully understand" should be " To fully understand".
5) Lines 108-109; "such as clustering algorithms and dimensionality reduction algorithms" should be "such as clustering and dimensionality reduction algorithms".
6) In the Figure 1 legend, please define what is "Residual samples".
7) Lines 153-156; Please use a different numbering scheme (other than 1,2,3,4) for the exclusion criteria. Also, in line 155, number 4 has been used two times.
8) In the supplementary file S3_standard_function and Figure 2 legend, " Characteristics and formulas of 23 standard functions", Is it 23 or 23A? Throughout the manuscript, the authors have used the term 23 A. If it is 23A then in line 197 "23A common test function" should be "23A common test function ( supplementary file; S3_standard_function.docx).
9) Line 254-255; Cite the reference for "t-test" and "t-test x2 test".
10) Line 263; "between groups, namely, age, hypertension grade, MuLBSTA (Iijima et al., 2021)," - It is not clear what is "namely" here. If it is not a variable then please rephrase the sentence accordingly in the manuscript. 
11) Lines 265-267; Please cite the reference to support the MuLBSTA score statements.
12) Line 304; "COVID-19 nucleic acid detection (O gene)". Is it COVID-19 nucleic acid detection or COVID-19 nucleic acid detection test? Is it not clear what is O gene here? To make it clear to the readers please rephrase this sentence in the manuscript. 
13) Line 367-368; Is it "ACE inhibitors/ARBs" or "ACEIs/ARBs"? Please correct it accordingly in the manuscript.

·

Basic reporting

Please discuss clearly why you did not use the traditional supervised machine learning method.

Experimental design

No comments.

Validity of the findings

1. Why such a less patient number? The statistics show that none of the p values is significant (Table 2). Would you please explain this more? In the text Line No 284, 296, and 298 are written differently.

2. Line 296-298: “Second, for the clinical symptom indices, the differences in the clinical symptom indices between the two groups of patients were small and not statistically significant (P>0.05)”. Please paraphrase this sentence.

Additional comments

No comments.

·

Basic reporting

Authors have addressed the comments in the previous review. The manuscript does not require any further revisions and is acceptable as is.

Experimental design

'no comment'

Validity of the findings

'no comment'

Reviewer 4 ·

Basic reporting

The authors performed a prognostic evaluation using unsupervised learning in the treatment of COVID-19.

Experimental design

The methods section of this paper is clear and nicely written.

Validity of the findings

- Figure 2 has too many subplots. The readers can rarely see all the subplots and therefore get any meaningful information from them.

- In Figure 5: feature number, do you have any meaningful description for each feature number? Otherwise, the plot is not informative.

---

## Round 0.4 · Minor Revisions

Thank you for your careful revision. I recommend the acceptance of this manuscript after a few minor changes. Please see below:

Minor comments:

1. Please include the GitHub link in the abstract. 
2. Line 16: Please correct "angiotensin-converting enzyme (ACEI)/". ACEI stands for Angiotensin-converting enzyme inhibitors. 
3. The authors used two full forms for ARB i.e., in lines 16-17; angiotensin receptor B (ARB), whereas in line 55, angiotensin II receptor blockers (ARBs). Please make sure that all the full forms are accurate and consistent throughout the manuscript. 
4. Line 68: ACEI/ARBs should be ACEI/ARB.
5. Line 71: ACEI/ARBon should be "ACEI/ARB on"
6. Line 389: "By blocking ACE2 receptors, ACE/ARB" - please check whether ACE/ARB should be ACEI/ARB or not. 
7. Line 476; "All members of the xxx Study" - what is xxx? Please correct it accordingly.
8. Table 3: Please check and correct accordingly "ACEII/ARB" or "ACEI/ARB"

Previous comments not answered: 

1. Line 306; "COVID-19 nucleic acid detection (O gene)". Is it COVID-19 nucleic acid detection or COVID-19 nucleic acid detection test? Is it not clear what is O gene here? To make it clear to the readers, please rephrase this sentence in the manuscript.  
2. In the Figure 1 legend, please define what is "Residual samples" - There are no figure legends in the main manuscript.

·

Basic reporting

Good. No further edits.

Experimental design

Good. No further edits.

Validity of the findings

Good. No further edits.

Additional comments

Photos need more resolution.
Many spaces have no gaps between words and brackets '().'

---

## Round 0.5 · Minor Revisions

Thank you for your diligent efforts in revising the manuscript. The authors have carefully considered all the issues raised by both myself and the reviewers. I suggest that the manuscript is ready for publication. However, to ensure the final version meets all requirements, please address the minor corrections listed below before submitting it.

1) Previous comment not answer: The authors used two full forms for ARB i.e., in lines 16-17; angiotensin receptor B (ARB), whereas in line 55, angiotensin II receptor blockers (ARBs). Please make sure that all the full forms are accurate and consistent throughout the manuscript.

2) There is a typo error (i.e., Table 7) listed in Table3.

---

## Round 0.6 · accepted · Accept

The authors have carefully considered and included all the suggestions and comments provided by both myself and the reviewers.